# A Novel Cold-Adapted Nitronate Monooxygenase from *Psychrobacter* sp. ANT206: Identification, Characterization and Degradation of 2-Nitropropane at Low Temperature

**DOI:** 10.3390/microorganisms12102100

**Published:** 2024-10-21

**Authors:** Yatong Wang, Shumiao Hou, Qi Zhang, Yanhua Hou, Quanfu Wang

**Affiliations:** 1School of Marine Science and Technology, Harbin Institute of Technology, Weihai 264209, China; wyt-hit@hit.edu.cn (Y.W.); 23S030086@stu.hit.edu.cn (S.H.); 13614511315@163.com (Q.Z.); 2School of Environment, Harbin Institute of Technology, Harbin 150090, China

**Keywords:** nitronate monooxygenase, cold-adapted, nitroalkane compounds, biodegradation

## Abstract

Aliphatic nitro compounds cause environmental pollution by being discharged into water with industrial waste. Biodegradation needs to be further explored as a green and pollution-free method of environmental remediation. In this study, we successfully cloned a novel nitronate monooxygenase gene (*psnmo*) from the genomic DNA library of *Psychrobacter* sp. ANT206 and investigated its ability to degrade 2-nitropropane (2-NP). Homology modeling demonstrated that PsNMO had a typical I nitronate monooxygenase catalytic site and cold-adapted structural features, such as few hydrogen bonds. The specific activity of purified recombinant PsNMO (rPsNMO) was 97.34 U/mg, rPsNMO exhibited thermal instability and reached maximum catalytic activity at 30 °C. Moreover, rPsNMO was most active in 1.5 M NaCl and remained at 104% of its full activity in 4.0 M NaCl, demonstrating its significant salt tolerance. Based on this finding, a novel bacterial cold-adapted enzyme was obtained in this work. Furthermore, rPsNMO protected *E. coli* BL21 (DE3)/pET28a(+) from the toxic effects of 2-NP at 30 °C because the 2-NP degradation rate reached 96.1% at 3 h and the final product was acetone. These results provide a reliable theoretical basis for the low-temperature degradation of 2-NP by NMO.

## 1. Introduction

Nitro compounds are a typical class of nitrogen-containing organic compounds, classified into two groups according to their different structures: aliphatic and aromatic nitro compounds. Some nitroalkanes are carcinogenic to rats [1,2,3]. Unfortunately, nitro compounds are found in industrial waste discharges and ozone-treated sewage [4,5], leading to the unavoidable exposure of humans and the natural environment to these substances. More seriously, human exploration and animal activities have brought these organic pollutants to the Antarctic [6,7], causing environmental pollution and ecological damage. The nitro group in nitroalkanes has a unique affinity for electrons, which makes it stable [8]. However, some microorganisms can survive under these extremely harsh conditions; thus, this extreme environment may provide potential biological resources for the degradation of nitroalkanes. Physical and chemical degradation methods have been extensively explored to convert nitroalkanes into less toxic substances. Although the degradation effect was remarkable, it was limited by complex operation procedures, expensive equipment and other problems [9]. Considering these factors, the use of biocatalysis to degrade nitroalkanes in the environment has become a hot research topic. A strain of *Geobacillus thermoglucosidasius* W-2 expressed an enzyme capable of degrading nitroalkanes [10]. Previously, some enzymes were also found to be able to degrade nitroalkanes, such as sulfotransferase(s) [11], nitroalkane oxidase [12] and nitronate monooxygenase [13].

Among the potential enzymes for nitroalkanes degradation, nitronate monooxygenase (NMO, EC 1.13.12.16), formerly known as 2-nitropropane dioxygenase (EC 1.13.11.32), was a flavonoid protein that efficiently catabolizes and metabolizes nitroalkanes [14]. It catalyzes different types of substrates (types I and II). Type I NMO has four highly conserved motifs and can effectively oxidize anionic alkyl nitrates. Type II NMO can catalyze anionic alkyl nitrates and neutral nitroalkanes. In addition to four completely conserved amino acid residues, type I NMO also included a tyrosine residue, which was replaced with histidine in only one-quarter of the sequences [15]. A strain of *Geobacillus thermophilus* was discovered in northern China. They extracted and purified the alkane monooxygenase expressed by the strain and studied its degradability [16]. Furthermore, a prokaryotic NMO from *Pseudomonas aeruginosa* PAO1 (PaNMO) [17] and a eukaryotic NMO from *Cyberlindnera saturnus* (CsNMO) [18] were biochemically characterized. The isolation and purification of cold NMO and its properties still require further development. Cold-adapted enzymes exhibit higher conversion rates, enhanced substrate and product specificity, and the production of fewer by-products, making them essential for modern industrial applications [19]. Moreover, recent studies have isolated several cold-adapted enzymes from microorganisms in the Antarctic sea-ice, which are now widely utilized in various industrial sectors due to their outstanding biochemical properties [20,21]. Considering these aspects, we needed to further study NMO expressed by Antarctic *Psychrobacter*.

In this study, the *psnmo* gene was found in the *Psychrobacter* sp. ANT206, which was isolated from Antarctic sea-ice. The *psnmo* gene was cloned, and the protein expressed in *E. coli* was purified. Subsequently, the effects of temperature, pH, metal ions and other factors on the activity of the purified recombinant PsNMO (rPsNMO) enzyme were systematically investigated. In addition, the protective effect of NMO against *E. coli* at low temperatures was analyzed. The degradation of nitroalkanes by this enzyme was further evaluated in detail. Therefore, this study suggests that NMO may effectively treat 2-NP in low-temperature environments.

## 2. Materials and Methods

### 2.1. Gene Cloning and Sequence Analysis of Psnmo

The Primer Premier 5.0 software was used to design primers for PCR amplification according to the *psnmo* sequence and annotation in ANT206’s whole-genome sequence. The full-length *psnmo* was amplified with 5′-CCCTCTAGAATGACCTTACTGAACA-3′ (*Xba*I site) and reverse primer 5′-ACTGAATTCCTTACCAGTCGCTCTA-3′ with 5′ (*EcoR*I site) via PCR. The sequence after *psnmo* sequencing was uploaded to the database (GenBank accession: OQ747184). The complete amino acid sequence of *psnmo* was analyzed using an open reading frame (ORF) finder to predict the protein’s molecular weight. NMO genes from different sources were selected, and multiple sequence alignment analysis of the target gene was performed using Bioedit 7.0 and the ESPript 3.0 program (https://espript.ibcp.fr/ESPript/cgi-bin/ESPript.cgi, last accessed on: 17 October 2024).

### 2.2. Bacterial Strains, Plasmid Construction, and Growth Conditions

The culture of the ANT206 strain was carried out according to the method previously described [22]. Then, the target gene *psnmo* was digested with *Xba*I and *EcoR*I, and the amplified *psnmo* gene was ligated with pET28a(+) to obtain the expression vector. *E. coli* BL21 (DE3) was used as the host strain for DNA manipulations and the overexpression of PsNMO, and was cultured in LB medium containing kanamycin (100 mg/L) for 12 h at 37 °C with shaking.

### 2.3. Homology Modeling and Structural Analysis of PsNMO

The PsNMO structure was constructed with AlphaFold3 (https://alphafoldserver.com/, last accessed on: 17 October 2024). The PyMOL 2.2.0 software was used to obtain the visual protein structure at the amino acid level [23]. Protein structural parameters such as hydrogen bonding and salt bridges were analyzed utilizing an electrostatic interaction calculator program (https://proteintools.uni-bayreuth.de/, last accessed on: 17 October 2024). Furthermore, the evolutionary relationships between different species were investigated by constructing a phylogenetic tree with the MEGA 11.0 software and TVBOT (https://www.chiplot.online/tvbot.html, last accessed on: 17 October 2024), which incorporated the PsNMO protein sequences.

### 2.4. Protein Expression and Purification of PsNMO

An amount of 50 μL of recombinant *E. coli* BL21 (DE3)/pET28a(+), named *E. coli* BL21 (DE3)/pET28a(+)-rPsNMO, was added to 5 mL of a liquid medium with kanamycin at 37 °C for overnight activation culture. The fermentation was then transferred to 100 mL of a liquid medium, and 1.0 mM isopropyl β-D-thiogalactoside (IPTG) was subsequently added to induce for 16 h at 15 °C when the cell density at 600 nm (OD_600_) reached 0.6–0.8. The recombinant bacteria were washed twice with PBS (pH 8.0) and centrifuged at 10,050× *g* for 7 min to harvest the precipitate. The cells were broken at a low temperature for 12 min in the lysis buffer (PBS, 0.5 M Tris-HCl). After centrifugation, the pellet was washed with PBS at 10,050× *g* to remove cell debris, followed by 8 M urea to lyse the inclusion bodies. Finally, the crude extract (inclusion bodies solubilized in 8M urea) of PsNMO in the supernatant was incubated at 4 °C for 1 h and then centrifuged.

The purification of the His-tagged (N-ter) rPsNMO was achieved through Ni-NTA affinity chromatography. The resulting purified protein was eluted using imidazole buffers at concentrations of 10, 50, 100 and 250 mM, in a 20 mM Tris-HCl, 500 mM NaCl solution at pH 8.0, with a flow rate of 1.0 mL/min. Finally, the purified PsNMO protein was collected via 50 mM imidazole elution and stored at −20 °C. The purification fold was obtained by the ratio of the specific activity of purified PsNMO to that of the crude enzyme. SDS-PAGE was conducted using 12.5% polyacrylamide gels and the molecular weight of the protein band was determined by constructing a standard curve from known molecular weight markers and calculating the relative mobility of the bands based on their migration distance relative to the dye front. Using bovine serum albumin (BSA) as a standard, the Coomassie Brilliant Blue method was employed to quantify the total protein (mg). The origin of the protein marker used for the SDS-PAGE gel was purchased from Takara (Dalian, China; code no. 3450Q).

### 2.5. Enzymatic Assays of rPsNMO

NMO can degrade 2-NP in the environment, so 2-NP was used as a substrate to evaluate the activity of this enzyme [24,25,26]. An equal amount of potassium hydroxide was slowly added, and the equilibrium substrate was incubated for 24 h to obtain 2-NP in its anionic form. The degradation of 2-NP by NMO produces nitrite, which reacts with the chromogenic agent to produce a purplish-red substance Therefore, the sodium nitrite standard curve was used to measure the content of nitrite produced in the reaction. This experiment was a modified version of the previous method for determining the activity of NMO [24], detailed as follows: The enzyme activity was determined using spectrophotometry (UV2000, Shimazu, Kyoto, Japan). The substrate with a concentration of 10 mM and a potassium phosphate buffer solution with pH 8.0 were mixed into the reaction system, and 10 μL of the purified enzyme solution was added for 5 min at room temperature. An amount of 10 μL of a glacial acetic acid terminator was subsequently added to terminate the reaction. An amount of 50 μL of sulfonamide and an equal volume of α-naphthalene chromogenic reagent were added into the reaction system and mixed. After reacting in the dark at room temperature for 10 min, the reaction was measured at 540 nm. The measured absorbance value was substituted into the standard curve, and the NMO enzyme activity was calculated using the enzyme activity Formula (1). The enzyme activity was defined as the amount of the enzyme required to catalyze the generation of 1 μmol of nitrite within 1 min as 1 unit.
U = ΔCVN/V_E_T(1)
where U is the enzyme activity of rPsNMO, ΔC is the nitrite content produced, V is the total volume of the reaction, N is the dilution multiple of the enzyme, V_E_ is the volume of the enzyme used in the reaction process and T is the reaction time.

### 2.6. Study on Enzymatic Characteristics of rPsNMO

#### 2.6.1. Effect of Temperature on Enzyme Activity

In the temperature range of 0–60 °C, the activity of rPsNMO with different temperature gradients was determined with a gradient of 5 °C to determine the optimal temperature for the catalytic effect of rPsNMO. The enzyme and substrate were incubated at different temperatures (30, 40, 50 and 60 °C) for a period of time (10 min, 20 min, 30 min, 40 min, 50 min, 60 min and 70 min), and the residual enzyme activity was measured as described in Section 2.5.

#### 2.6.2. Effect of Buffer pH on rPsNMO Activity

Substrates were added to different buffers to determine the optimal pH at which rPsNMO functions, including a 0.5 M dipotassium citrate-potassium hydrogen phosphate buffer (citric acid-K_2_HPO_4_; pH 5.0–7.5) and a dipotassium-potassium dihydrogen phosphate buffer (KH_2_PO_4_- K_2_HPO_4_; pH 7.5–9.5). The sample solution was incubated for 5 min at 30 °C. The samples were incubated under the above conditions for 30 min to determine the stability of rPsNMO in the buffer solutions with different pH values (5.0–9.5).

#### 2.6.3. Effects of Salt Concentration and Different Reagents on rPsNMO Activity

The purified rPsNMO was incubated in 0–4.0 M NaCl at 30 °C for 30 min, and the remaining activity was assayed with the standard enzyme assays. The effects of different reagents on the rPsNMO activity were assayed with the standard enzyme assay after pre-incubating the enzyme in different metal ions and chemical reagents at 30 °C for 30 min. The enzyme activity assayed without any reagent was defined as the control (100%).

#### 2.6.4. rPsNMO’s Thermodynamic and Kinetic Parameters

The kinetic and thermodynamic properties were exercised in different concentrations of 2-NP solution (6.0, 7.0, 8.0, 9.0, 10.0, 11.0 and 12.0 mM) in potassium dihydrogen phosphate/dimethyl hydrogen phosphate as the buffer solution (pH 7.5) at different temperatures (0 °C, 10 °C, 20 °C and 30 °C). The activity estimation was conducted as previously described. The Lineweaver–Burk plot method was used to calculate the *K*_m_ and *V*_m_ of rPsNMO to assess the kinetics parameters [27]. The *k*_cat_ and thermodynamic parameters (Δ*H*, Δ*G* and Δ*S*) were obtained using the results of the kinetic calculations and modification methods [28].

#### 2.6.5. Statistical Analysis

The maximum enzyme activity of each group was normalized to 100% for descriptive statistical analysis. Every experiment was repeated at least 3 times, and the values were the means of triplicate samples from a typical experiment. The error bars represent the standard errors of the means.

### 2.7. Protection of E. coli from 2-NP Toxicity by rPsNMO

The activated recombinant *E. coli* BL21 (DE3) harboring the pET28a(+)-*psnmo* vector was inoculated into a liquid medium containing 10 mM 2-NP. The enzyme is produced within the inclusion body, so it is not secreted into the extracellular medium. Therefore, 1 mg of the purified enzyme solution was additionally inoculated into the medium, and the medium without the enzyme solution was incubated at 30 °C as a control to ensure the enzyme functioned at the optimal temperature. The strains were added to the fresh LB media until OD_600_ achieved 0.05. Then, the strains were incubated at 30 °C for 12 h. The OD_600_ values of the strains were determined via spectrophotometry (UV2000, Shimazu, Kyoto, Japan) at 600 nm every hour, and the growth curves of the strains in the experimental and control groups were observed.

### 2.8. Degradation Efficiency Analysis of 2-NP via HPLC

Liquid chromatography (LC) techniques are highly effective for the structural elucidation and quantification of degradation products and impurities, due to their exceptional resolution and sensitivity [29]. High-performance liquid chromatography (HPLC) was employed for precise measurement to accurately quantify the residual 2-NP potentially degraded by the Section 2.7 enzymes [30]. The HPLC setup and conditions were as follows: The HPLC system (Prominence HPLC, Shimadzu) was configured with a C18 column (250 mm × 4.6 mm; 5 μm particle size) for the optimal separation of the compounds. The mobile phase consisted of methanol and water in a 60:40 (*v*/*v*) ratio. A flow rate of 0.5 mL/min was maintained, and the column temperature was set at 30 °C to ensure optimal elution. The detection wavelength was set to 280 nm, which aligns with the maximum absorbance of 2-NP, ensuring high detection sensitivity [31]. A sample injection volume of 10 μL was used for each analysis. Descriptive statistical analysis was performed on the experimental results, which were repeated in triplicate. The standard deviations are represented as error bars.

### 2.9. Extraction and Qualitative Analysis of Degradation Products via GC-MS

The degradation products were obtained via ultrasonic extraction (KQ5200QE, 5 min, 30 °C, 40 Hz) [32,33]. The reaction system consisted of 4 mL of a phosphate buffer solution with pH 7.5, 4 mL of the purified enzyme solution and 10 mM 2-NP at 30 °C. After three hours of reacting in the dark at 120× *g*, glacial acetic acid was added to the solution to terminate the reaction. Then, ethyl acetate was added and the degradation products were enriched and condensed via ultrasonic extraction for 30 min under ice bath conditions. The supernatant was centrifuged at 7500× *g* for 3 min, filtered via an ultrafiltration membrane with a diameter of 0.22 μm and stored for later use. The stored samples were qualitatively analyzed using a Thermo Fisher GC device. The collected samples were analyzed using a Shimadzu GC-MS (QP2010, Kyoto, Japan). The detection conditions were as follows: a capillary column was used, and the temperature at the inlet was 200 °C, the GC-MS interface temperature was 220 °C; the ion source temperature was 230; and the ion scanning range was 28–300 m/z. The column heating procedure was as follows: the initial temperature was 40 °C and held for 5 min, and the temperature was increased by 20 °C per minute to 230 and maintained for 2 min. Helium was used as the carrier gas. The full scanning mode was used for the samples.

## 3. Results and Discussion

### 3.1. Bioinformatics Analysis of rPsNMO

The *psnmo* sequence was 1044 bp long and encoded a protein containing 347 amino acids, while the NMO of *H. mrakii* contained 374 amino acid residues [34]. And the predicted molecular weight was 37.7 kDa. Thirteen α-helices and eleven β-folds in the tertiary configuration of PsNMO protein were identified (Figure 1a). The multiple sequence alignment results revealed that this sequence contained four conserved motifs, and the active sites were residues Met18, Asn67, His176, Tyr297 and Lys305, which were identified as class I NMO (Figure 1a). Furthermore, the phylogenetic tree results further identified the type of PsNMO as class I NMO (Figure 1b).

### 3.2. Homology Modeling and Structural Analysis of PsNMO

The results obtained using PyMOL as a protein visualization tool demonstrated a high degree of overlap between the catalytic sites of PsNMO and its homologous mesothermal enzyme from *Pseudomonas aeruginosa* (PaNMO, PDB ID: 4Q4K [17]) in their 3D structures (Figure 2). The PsNMO structure contained fewer hydrogen bonds than the homologous mesothermal PaNMO structure (Table 1), resulting in the decreased stability and thermal resistance of cold-adapted proteins [35], which have been observed in other cold-adapted enzymes such as laccase [20] and carbonic anhydrase [36]. PsNMO contained only half the amount of Arg residues in PaNMO, implying that less arginine can reduce the stability of the protein conformation. Furthermore, the PsNMO had a lower Arg/(Arg + Lys) ratio of 0.54 compared to the 0.83 ratio of PaNMO. This difference could be attributed to the elevated conformational flexibility of PsNMO, which was anticipated to enhance its catalytic activity at lower temperatures [37]. In summary, PsNMO had unique cold-adapted structural features to maintain its biological activity at low temperatures.

### 3.3. Expression, Purification, and Enzyme Assays of PsNMO

After *psnmo* was expressed by the recombinant *E. coli*, the NMO protein was purified via Ni-NTA. A single band (~38.5 kDa) appeared in the third lane of the SDS-PAGE profile (Figure 3). The result agreed with a previous work, in which 37 kDa was measured for PaNMO [17]. The specific activity of the purified rPsNMO was 97.34 U/mg, which was 14-fold higher than that of the protein crude extract. The protein recovery yield reached 51.31% (Appendix A). Meanwhile, the specific activity of NcNMO from *E. coli* was 184 U/mg [38].

### 3.4. rPsNMO’s Biochemical Properties

#### 3.4.1. Effects of Temperature and pH on Enzyme Activity

The enzyme activity of rPsNMO first increased and then decreased with the increasing temperature in the range of 0–60 °C. The enzyme activity was approximately 13.5% at 0 °C and reached the maximum when the temperature rose to 30 °C (Figure 4a). The highest activity of cold-adapted bacterial laccase [20] also occurred at 30 °C, which was higher than that of cold-adapted carbonic anhydrase from the psychrophilic bacteria *Colwellia* sp. NJ256 (25 °C) [36]. The enzyme activity significantly decreased after 50 °C (Figure 4b), and the residual activity was only 4% at 60 °C, indicating that the enzyme was sensitive to high temperatures. In terms of the enzyme’s thermal stability, the enzyme activity decreased by approximately 61% when incubated at 60 °C for 10 min, while the NMO from *Geobacillus thermoglucosidasius* W-2 retained better activity in the range of 50–70 °C [15]. The above results further indicated that the enzyme was cold-adapted and confirmed its thermal instability [39]. Figure 4c showed an increasing trend from 5.0 to 7.5 and then a decreasing trend from 7.5 to 9.5, indicating that pH 7.5 was the optimal pH. Moreover, the highest activity of NMO from *Pseudomonas aeruginosa* PAO1 presented at pH 7.5 [17]. Existing studies on the properties of NMO stated that the optimum pH for the catalytic action of NMO was 7.4 [13,38]. In addition, the enzyme illustrated high activity in the pH range of 7.5–8.5 (Figure 4d), indicating that the enzyme had excellent catalytic activity and stability under weak alkaline conditions.

The highest enzyme activity of approximately 148% was observed at a NaCl concentration of 1.5 M (Figure 4e). The enzyme activity decreased to approximately 104% as the salt concentration continued increasing to 4.0 M. In general, this study found that the enzyme could still maintain high activity in NaCl concentrations of 0–4.0 M. These data indicated that rPsNMO had excellent salt tolerance. This phenomenon was also observed in the nitroreductases and glutathione S-transferases of Antarctic sea-ice microorganisms [21,40]. It was speculated that this result might be related to the high-salinity environment in which Antarctic microbes live. The extensive discharge of organic saline wastewater containing nitroalkanes has long been a persistent and escalating problem in industries, such as printing, coal chemical processing and pharmaceuticals [41]. Physicochemical treatments are often costly and complex, require frequent maintenance, and struggle to remove organic matter. Consequently, low-cost, easy-to-operate biological treatment technologies are becoming a research focus. Therefore, the high salt tolerance of rPsNMO shows potential for future applications in the treatment of high-salinity organic wastewater.

#### 3.4.2. Effects of Different Reagents on rPsNMO Activity

EDTA at a concentration of 1 mM had a minimal inhibitory effect on the rPsNMO enzyme activity (Table 2) and the NMO activity from *Neurospora crassa* [42]. In addition, the inhibition of rPsNMO with Cu^2+^ concentrations of 1 mM and 5 mM was 55.4% and 25.2%, respectively. The NMO from *Streptomyces ansochromogenes* was also strongly inhibited by Cu^2+^ [43]. 2-mercaptoethanol had a strong inhibitory effect on the rPsNMO enzyme activity. The enzyme activity decreased to 34% with a 2-mercaptoethanol concentration of 1 mM. The residual enzyme activity was only 23% when 1 mM 2-mercaptoethanol was added to the nitroalkane oxidase from *Streptomyces* [44]. Therefore, 2- mercaptoethanol significantly influenced enzyme activity. Mg^2+^ promoted enzyme catalysis at low concentrations, and the rPsNMO activity increased by 139.8% at 1 mM.

### 3.5. Kinetic and Thermodynamics Parameter Measurements of rPsNMO

Double inverse curves were plotted after the reaction of the enzyme with different concentrations of 2-NP at 0–30 °C (Figure 5). The kinetic parameters *V*_m_ and *K*_m_ were calculated according to the curve intercept. The results reflected the gradual increase in *k*_cat_ with increasing temperature (Table 3). In general, the cold-adapted enzymes were highly efficient in compensating for the reduced reaction rates at low temperatures by improving the *k*_cat_ value. Therefore, it was postulated that rPsNMO may adapt to low temperatures by increasing *k*_cat_. It was evident that as the temperature increased, and the *k*_cat_ value exhibited a gradual increase, a trend comparable to the observed behavior of cold-adapted laccase at temperatures between 10 and 30 °C [20]. The catalytic efficiency (*k*_cat_/*K*_m_ = 521.44 M ^−1^ S ^−1^) was highest at 30 °C, while the catalytic efficiency of rNcNMO for 2-NP at 30 °C was 250 M ^−1^ S ^−1^ [45]. Δ*H* decreased with increasing temperature, exhibiting the lowest value at 30 °C (Table 3). This weakened the exponential dependence of the reaction rate on the temperature [46], which promoted the binding of the enzyme to the substrate, and the catalytic effect was improved.

### 3.6. Protection of E. coli from 2-NP Toxicity by rPsNMO

The transformed *E. coli* BL21 (DE3)/pET28a(+)-rPsNMO and *E. coli* BL21 (DE3)/pET28a(+) were cultured in a liquid culture containing 2-NP, and the growth of the OD_600_ was continuously monitored. The *E. coli* BL21 (DE3)/pET28a(+) inoculated with 1 mg of rPsNMO grew normally in the presence of 10 mM 2-NP (Figure 6). Additionally, the growth of *E. coli* that had not been inoculated with the enzyme was inhibited. A previous study illustrated that nitroalkane pollutants completely inhibited the growth of *E. coli*, resembling the results of this experiment [13]. The results clearly reflect toxicity of 2-NP against *E. coli* BL21 (DE3)/pET28a(+) and the detoxification effect of rPsNMO on *E. coli* BL21 (DE3)/pET28a(+).

### 3.7. Degradation Efficiency Analysis of 2-NP via HPLC

This experiment used HPLC to monitor the residual amount of 2-NP to further determine the degradation effect of rPsNMO on 2-NP. The experiment used HPLC to monitor the residual amount. The rapid degradation of 2-NP by rPsNMO, with 96.1% degraded within 3 h, highlights the enzyme’s remarkable efficiency compared with previous studies, where 80% degradation was achieved over 3 days [47]. 2-NP levels remained stable for the first hour (Figure 7), indicating a potential lag phase before optimal enzyme-substrate interactions occurred. Following this, the sharp decrease in the 2-NP concentration underscores the enzyme’s effective activity. This was confirmed by the control group, where 2-NP levels remained stable throughout the experiment. These findings suggest that rPsNMO would offer significant potential for the rapid bioremediation of 2-NP in contaminated environments.

### 3.8. Extraction and Qualitative Analysis of Degradation Products by GC-MS

The analysis was carried out with a temperament combination analyzer to determine the degradation of 2-NP by PsNMO. The molecular ion peak and fragment ion peak were 58.2 and 43.1, respectively. The fragment peak of 43.1 was obtained by the molecule losing a methyl molecule after being bombarded, as shown in Figure 8. The substance with a molecular weight of 58 mainly consisted of two compounds: acetone and propional. The spectrum was found to be closest to the acetone mass spectrum via comparison with the standard spectrum in the spectrum library. Therefore, 2-NP was degraded into acetone by PsNMO in this experiment. The degradation process is speculated to involve the oxidization of the anionic form of 2-nitropropane to generate a 2-nitropropane radical, which reacted with a superoxide anion to produce a peroxide intermediate. This intermediate then reacted with another anionic 2-nitropropane molecule, releasing two nitrite ions and forming two acetone molecules [42]. Other studies have also demonstrated that NMO degrades 2-NP into acetone [42,48]. Moreover, several other enzymes have been shown to convert 2-NP into acetone [3,43].

In conclusion, rPsNMO demonstrated significant activity in both low-temperature and high-salinity environments. Future studies could focus on biocatalysts that develop the cell surface display of PsNMO to extend its potential applications. This approach leverages anchoring motifs (carriers) and host systems to facilitate the functional expression of passenger proteins (target enzymes) on the bacterial outer membrane [20,36]. However, challenges related to low reusability may arise in practical applications; nevertheless, improving the reusability of whole-cell catalysts remains highly necessary. Metal-organic frameworks (MOFs), known for their large surface area, high pore volume, excellent crystallinity and tunable structure, have recently gained attention across various fields. Immobilizing whole-cell catalysts using MOFs, such as isoreticular metal-organic frameworks and zeolitic imidazolate frameworks, may enhance reusability while reducing catalytic activity loss, thereby supporting broader industrial applications. Consequently, this strategy is anticipated to enable cold-adapted PsNMO to effectively catalyze the enzymatic degradation of nitroalkanes under low-temperature and high-salinity conditions.

## 4. Conclusions

In the present work, a novel NMO gene from *Psychrobacter* sp. ANT206 was successfully cloned, expressed and characterized. After bioinformatic characterization, rPsNMO was found to exhibit structural features of cold-adapted enzymes, explained by the fewer hydrogen bonds and Arg/(Arg + Lys) in rPsNMO. Subsequently, the enzyme’s enzymatic properties were measured in vitro, and rPsNMO was identified as a cold-adapted enzyme with excellent salt tolerance. In addition, experiments indicated that the enzyme prevented the 2-NP-induced growth inhibition of *E. coli* and effectively converted 2-NP into acetone. Further research on PsNMO may alleviate the problem of 2-NP pollution in low-temperature environments.

## Figures and Tables

**Figure 1 microorganisms-12-02100-f001:**
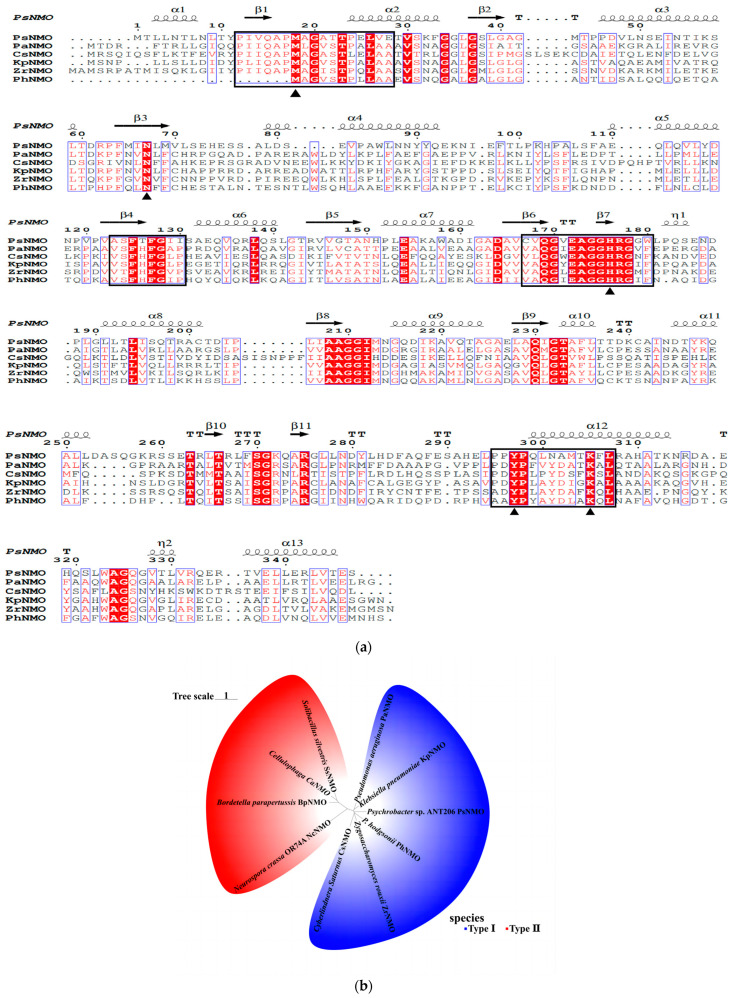
(**a**) Alignment of PsNMO and NMO sequences from other sources; (**b**) Phylogenetic trees of PsNMO. The species names and GenBank accession numbers were as follows: PsNMO, *Psychrobacter* sp. ANT206 NMO (OQ747184); PaNMO, *Pseudomonas aeruginosa* NMO (NP-252891.1); CsNMO, *Cyberlindnera saturnus* NMO (AAA64484.1); KpNMO, *Klebsiella pneumoniae* NMO (WP-004179795.1); ZrNMO, *Zygosaccharomyces rouxii* NMO (XP-002498653.1); PhNMO, *Pantholops hodgsonii* NMO (XP-005969806.1), CaNMO, *Cellulophaga* NMO (WP-013550013.1); BpNMO, *Bordetella parapertussis* NMO (WP-015040637.1); SsNMO, *Solibacillus silvestris* NMO (WP-014823224.1); NcNMO, *Neurospora crassa* OR74A NMO (XP-957588.1).

**Figure 2 microorganisms-12-02100-f002:**
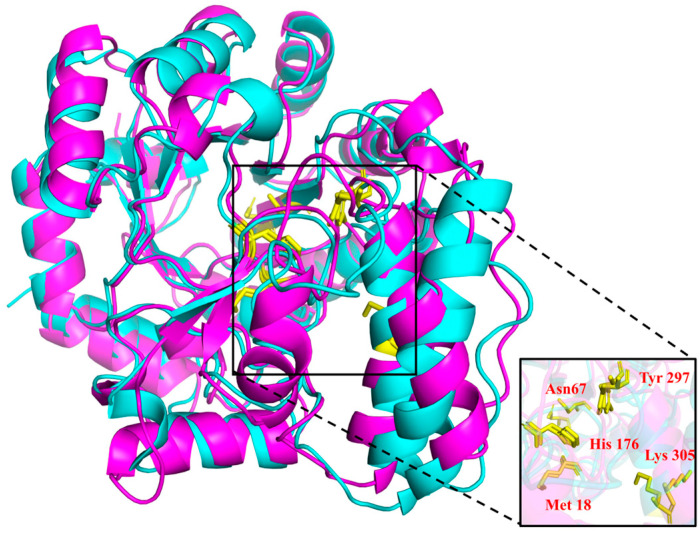
The 3D structure models of PsNMO (purple) and structure superimposition with PaNMO (blue). The catalytic triad residues were indicated as stick models.

**Figure 3 microorganisms-12-02100-f003:**
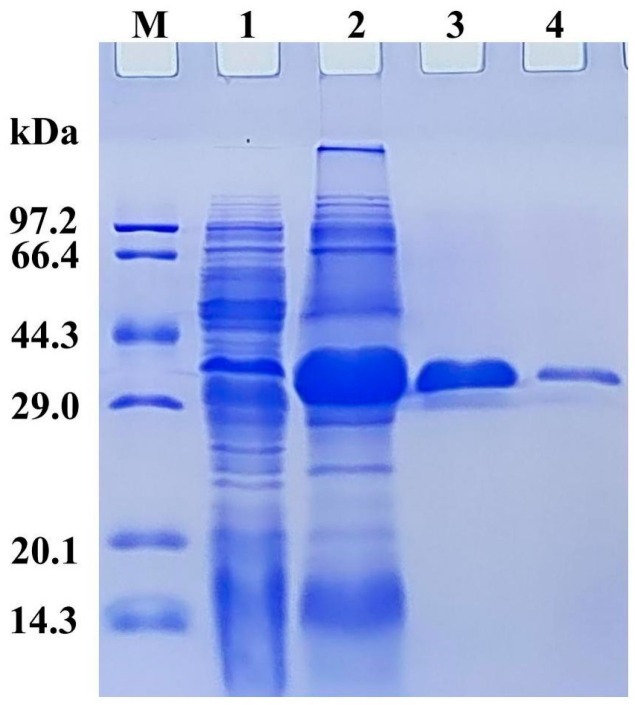
Expression and purification analysis of rPsNMO. M, molecular weight standard marker; 1, *E. coli* BL21 (DE3)/pET28a(+) crude extract; 2, crude extract from the BL21 (DE3)/pET28a(+)-rPsNMO with IPTG induction; 3, rPsNMO purified via Ni-NTA 50 mM imidazole elution; 4, rPsNMO purified via Ni-NTA 100 mM imidazole elution.

**Figure 4 microorganisms-12-02100-f004:**
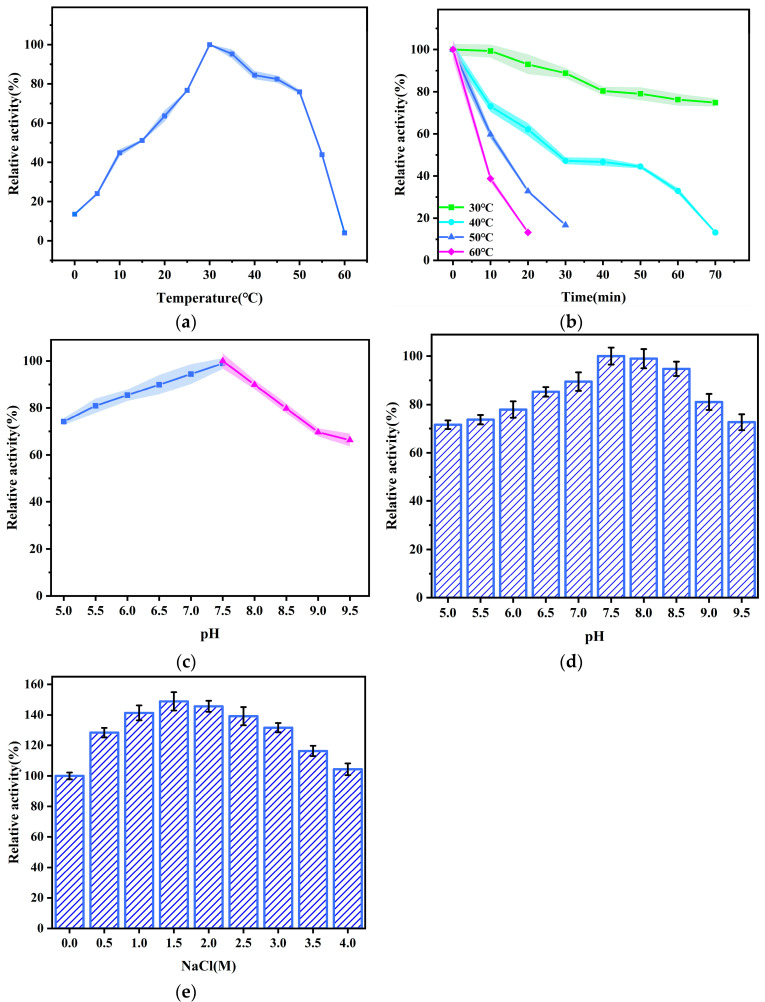
(**a**) Effect of temperature on activity at pH 7.5; (**b**) Effect of temperature on stability at pH 7.5; (**c**) Effect of pH on activity when incubated for 5 min at 30 °C, the buffers included citric acid-K_2_HPO_4_ (■) (pH 5.0–7.5), KH_2_PO_4_- K_2_HPO_4_ (▲) (pH 7.5–9.5); (**d**) Effect of pH on stability when incubated for 30 min at 30 °C. The maximum activity was set at 100%; (**e**) Effect of salt concentration on activity, the activity of enzyme treated with 0 M NaCl was defined as 100.0%.

**Figure 5 microorganisms-12-02100-f005:**
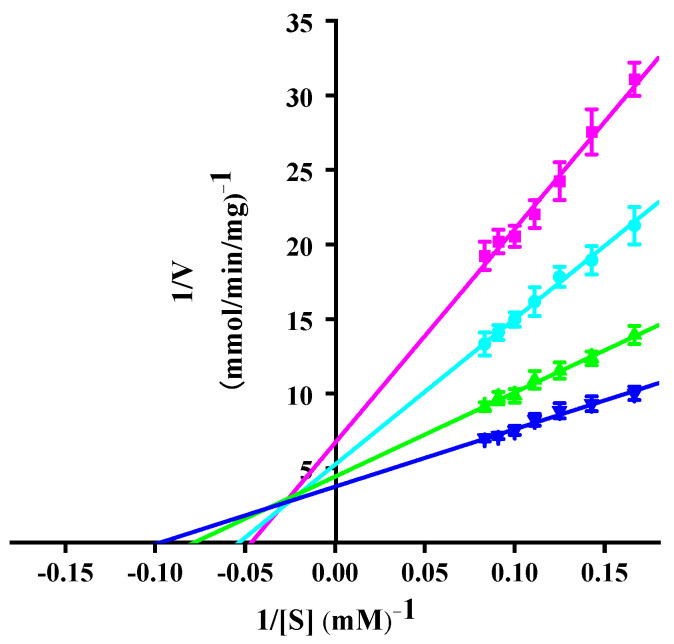
Double reciprocal Lineweaver-Burk plot of purified rPsNMO versus 2-NP; The temperatures for each curve were 0 °C (■); 10 °C (●); 20 °C (▲); and 30 °C (▼).

**Figure 6 microorganisms-12-02100-f006:**
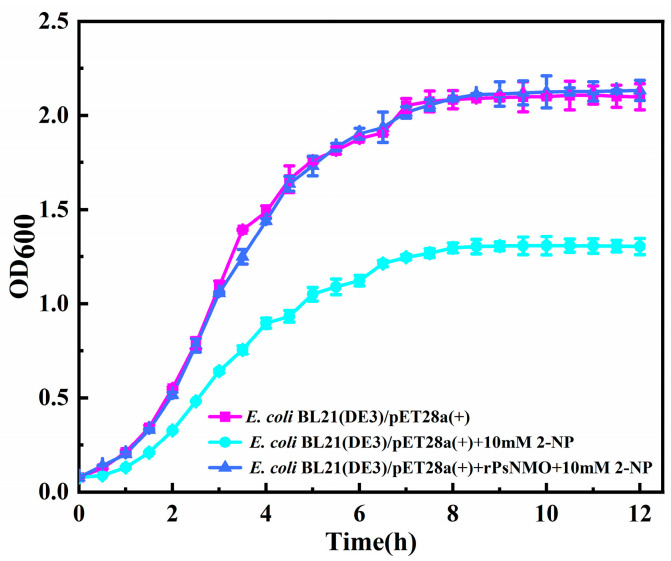
Growth curves of *E. coli* BL21 (DE3)/pET28a(+) and *E. coli* BL21 (DE3)/pET28a(+)-rPsNMO in LB medium containing 2-NP. The experimental group contained 1 mg of rPsNMO, and the control group did not contain the enzyme solution. The experimental group was co-cultured at a constant temperature of 30 °C for 12 h to ensure rPsNMO played a catalytic role.

**Figure 7 microorganisms-12-02100-f007:**
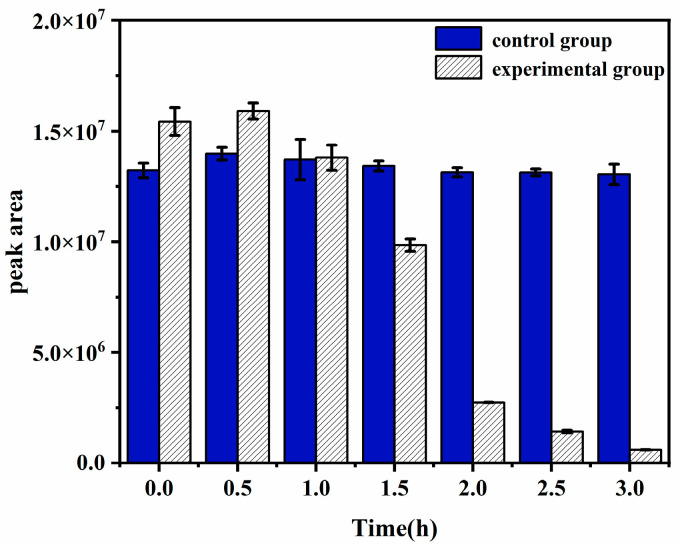
Degradation effect on 2-NP with enzyme solution. The initial concentration of 2-NP was 10 mM. The experiment was conducted at 30 °C; the results are presented in terms of the peak area. These results were analyzed using descriptive statistical methods, with the experiments performed in triplicate. The standard deviations are represented as error bars.

**Figure 8 microorganisms-12-02100-f008:**
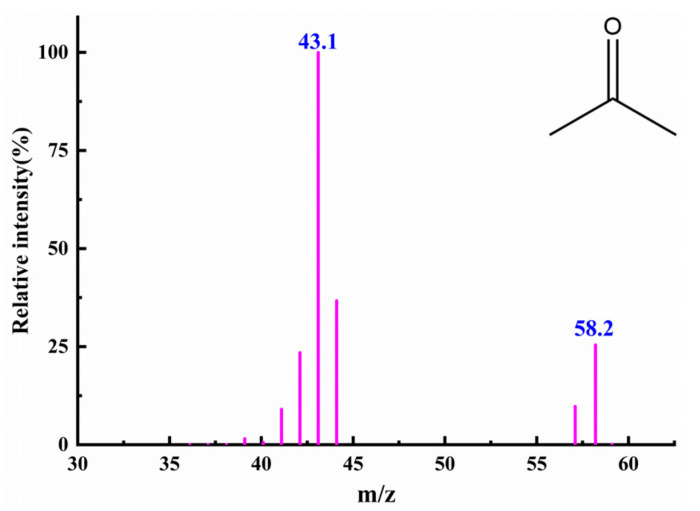
Material spectra of 2-NP final product degraded by rPsNMO.

**Table 1 microorganisms-12-02100-t001:** Comparison of structural adaption features between PsNMO and PaNMO (PDB ID: 4Q4K).

Parameters	PsNMO	PaNMO
salt bridge	9	7
Hydrogen Bonds	22	34
Glycine residues	26	34
Proline residues	18	26
Arginine residues	14	29
Arg/(Arg + Lys)	0.54	0.83

**Table 2 microorganisms-12-02100-t002:** Effects of different reagents on rPsNMO activity when incubated for 30 min at 30 °C, and 100% in terms of specific activity.

Metal Ions and Chemical Regents	Concentration	Relative Activity (%)	Concentration	Relative Activity (%)
None		100.0		100.0
Ag^+^	1 mM	95.1 ± 2.0	5 mM	127.2 ± 3.0
Ca^2+^	1 mM	121.4 ± 7.5	5 mM	95.2 ± 6.0
Cu^2+^	1 mM	55.4 ± 2.5	5 mM	25.2 ± 0.5
Mg^2+^	1 mM	139.8 ± 1.5	5 mM	112.6 ± 4.0
Fe^3+^	1 mM	103.9 ± 1.3	5 mM	87.4 ± 3.6
Ni^2+^	1 mM	104.9 ± 3.3	5 mM	85.4 ± 2.0
Cr^2+^	1 mM	91.3 ± 5.5	5 mM	99.0 ± 1.0
EDTA	1 mM	91.3 ± 4.3	5 mM	63.1 ± 2.7
2-mercaptoethanol	1 mM	34.0 ± 0.7	5 mM	0.0 ± 0.0

**Table 3 microorganisms-12-02100-t003:** Kinetic thermodynamic parameters of rPsNMO.

Temperature (°C)	*V*_m_ (mmol/min/mg)	*K*_m_ (mM)	*k*_cat_ (1/S)	Δ*H* (KJ/mol)	Δ*S* (J/mol.K)	Δ*G* (KJ/mol)
0	149.48	21.46	3.00	10.94	−194.96	64.16
10	191.21	18.69	3.84	10.86	−194.88	66.01
20	225.73	12.74	4.54	10.78	−195.46	68.05
30	265.96	10.26	5.35	10.69	−195.71	69.99

## Data Availability

The original contributions presented in the study are included in the article/Appendix A, further inquiries can be directed to the corresponding authors.

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
