# Peer review of "A Novel Cold-Adapted Nitronate Monooxygenase from Psychrobacter sp. ANT206: Identification, Characterization and Degradation of 2-Nitropropane at Low Temperature"

_microorganisms, 2024, doi:10.3390/microorganisms12102100_

Round 1
Reviewer 1 Report
Comments and Suggestions for Authors
The manuscript presents an interesting and relevant nitronate monooxygenase recombinant enzyme from an Antarctic psychrophilic bacterium (Psychrobacter sp. ANT206) for the biodegradation of 2-nitropropane (2-NP). However, there are several issues before it can be published in MDPI Microorganisms.
The first comment is related to English language, as the manuscript could greatly improve if is proofread by and expert technical native speaker.
- Abstract could be improved by including relevant results such as salt tolerance and optimal specific activity, please revise English.
- line 61, even though it appears in the abstract, please indicate the meaning of rPsNMO
- I believe that section 2.2 is prior to 2.1.
- are E. coli BL21 or BL21(DE3)? indicate the origin of the strain used.
- indicate if the his tag selected is N-ter or C-ter.
- please revise English in 2.4 as it is quite difficult to understand the methods for expression and purification of the recombinant enzyme.
- line 98, what is the composition of lysis buffer (or provider)?
- it is not clear from the text whether the crude extract corresponds to the enzyme expressed in the intracellular (soluble) fraction or refers to the inclusion bodies solubilized in 8M urea.
- Have the inclusion bodies undergone any refolding protocols?
- line 111 where was BSA used? please indicate the origin of the protein marker used for the SDS-PAGE gel shown in Figure 3.
- line 114, indicate the reference for the statement that NMO can degrade 2-NP in the environment.
- line 121, please revise Ultraviolet spectrophotometry? as you are using 540 nm. Please include the name, origin and country of the equipment used.
- line 130, add space between 1 and µM, use the symbol µ instead of u.
- line 137, how did you measure the residual activity?
- line 158, indicate how the Vm and Km values were actually obtained, method used, and the other parameters shown in Table 3: kcat, ΔH, ΔS, ΔG.
- line 161, is the E. coli used the one harboring the pET28a(+)-psnmo vector? indicate if is BL21 or BL21(DE3).
- line 166, indicate how the growth curves were determined (time and equipment used).
- Section 2.8 needs more details and improve English to properly understand, is this experiment related to 2.7? or not?
- line 176, 120 x g? what do you mean?
- line 182, Shimadzu, include the reference for the ultrasonic extraction method used.
- line 191, psnmo
- Figure 1a is quite difficult to observe the details, could be better to have it as supplementary in a full page.
- Figure 2 is also too small and the image quality is not very good (at least in the draft manuscript).
- Table 1 legend, include the codes and references used.
- line 225, how did you calculate the molecular weight of the single band shown? In the original SDS-PAGE submitted there is no dye front impeding to calculate the relative mobility of the protein bands. What is the predicted molecular weight from its sequence?. What is the protein marker used?. According to the original gel, are you sure lane 3 in Fig.3 corresponds to 50 mM imidazole? or could be 100 mM?.
- Please improve English in Section 3.4.1
- Figure 4 legend must improved to include relevant details. Fig 4a and b, please indicate the pH used, Fig 4c, d and e please indicate the temperature and time of incubation.
What is the Specific activity for each 100%?. What is the meaning of the error bars? how many experiments did you perform? what about statistical analysis of the data?. Please also include the information in the materials and methods sections.
- Table 2 legends, please also indicate the temperature and time of incubation and also 100% in terms of specific activity.
- Section 3.5, the determination of the kinetic constants was done by using Lineweaver-Burk, which even though it has been used in several publications, due to the double reciprocal, the experimental errors are largely amplified in comparison to other linearization methods. It would be better to fit the experimental data to other more suitable plots, such as Eadie-Hoffstee or Hanes-Woolf. Any method need to be detailed in the materials and methods also indicating the software used for the analysis of the data and obtaining of the parameters show in Table 3.
- Regarding the comparison of kinetics parameters with other enzymes it is always best to do it empirically due to the differences in the assays performed by different laboratories, conditions, concentration of enzyme and substrate, units definition, and method of determination, etc, or make sure that they are actually comparable in most of these terms.
- Table 3, how did you obtain kcat, ΔH, ΔS, ΔG?
- Section 3.6 it is not clear if the strain used was transformed or not and how much enzyme was added to the culture.
- Figure 6. Improve the figure legend by adding relevant information such as the amount of purified enzyme added to the culture (also indicate this in the materials and methods) and growth temperature.
- Section 3.7, it is not clear if this is a continuation from the previous experiment or a totally different one. Please include the information required to better understand and improve English to further discuss the obtained results.
- Figure 7. Please include the initial amount of 2-NP added in the legend and the conditions of the experiments and the calculation of the results shown. Statistical analysis of the data? how many replicates? meaning of error bars?
- Section 3.8, what is the level of certainty of this, Line 320 says it is speculated while in the abstract (line 20) seems very certain that the final product is acetone. What are the known routes for this enzyme? other reported degradation products? Please compare with other references to further discuss the results obtained.
- What are the projections and potential of the results shown? how this enzyme could be use in real life applications to actually biodegrade 2-NP?. Please elaborate to further enhance your discussion and conclusion.
- Table S1 seems incomplete, please also include the volume and protein concentration obtained in each step. what about errors?. it is not parameters for column 1, and is purification fold rather than multiples and yield rather than recovery.
- Table S2. Please indicate in the legend how these kinetic parameters were obtained (using the same enzyme assay and conditions? or not?) how are they comparable with your results?
Comments on the Quality of English Language
It needs to be improved, as it is a major flaw in this manuscript. Although it is fine in some sections, in others it seriously impairs the understanding of what the authors want to express. I strongly recommend finding a technical proofreading before re-submission to MDPI.
Author Response
The manuscript presents an interesting and relevant nitronate monooxygenase recombinant enzyme from an Antarctic psychrophilic bacterium (Psychrobacter sp. ANT206) for the biodegradation of 2-nitropropane (2-NP). However, there are several issues before it can be published in MDPI Microorganisms.
The first comment is related to English language, as the manuscript could greatly improve if is proofread by and expert technical native speaker.
Response: We consulted the Language Editing services on MDPI to check English (Order reference: english-edited-85849), and revised the language carefully, with the changes marked in Blue in the revised manuscript.
- Abstract could be improved by including relevant results such as salt tolerance and optimal specific activity, please revise English.
Response: We comprehensively considered your suggestion and another reviewer’s comment, the abstract has been modified. Please see page 1 line 16-19.
- Line 61, even though it appears in the abstract, please indicate the meaning of rPsNMO
Response: This part has been revised, please see page 2 line 67.
- I believe that section 2.2 is prior to 2.1.
Response: In this version, this part has been modified, please see page 2 line 73 and 84.
- Are E. coli BL21 or BL21(DE3)? indicate the origin of the strain used.
Response: This part has been modified, please see page 2 line 87-88; page 3 line 101-102; page 9 line 275.
- Indicate if the his tag selected is N-ter or C-ter.
Response: This part has been added, please see page 3 line 116.
- Please revise English in 2.4 as it is quite difficult to understand the methods for expression and purification of the recombinant enzyme.
Response: This part has been revised in this version, please see section 2.4.
- Line 98, what is the composition of lysis buffer (or provider)?
Response: This part has been added, please see page 3 line 108.
- It is not clear from the text whether the crude extract corresponds to the enzyme expressed in the intracellular (soluble) fraction or refers to the inclusion bodies solubilized in 8M urea.
Response: This part has been revised in this version, please see page 3 line 110.
- Have the inclusion bodies undergone any refolding protocols?
Response: The inclusion bodies did not undergo any refolding protocols.
- Line 111 where was BSA used? Please indicate the origin of the protein marker used for the SDS-PAGE gel shown in Figure 3.
Response: BSA is used as a standard to calculate the protein content. The origin of the protein marker has been added in new version, please see page 3 line 120-126.
- Line 114, indicate the reference for the statement that NMO can degrade 2-NP in the environment.
Response: We highly agree and appreciate very much for the Reviewer’s nice comments. The references had been added, please see page 3 line 129.
- Line 121, please revise Ultraviolet spectrophotometry? as you are using 540 nm. Please include the name, origin and country of the equipment used.
Response: This part has been modified, please see page 3 line 135-136.
- Line 130, add space between 1 and µM, use the symbol µ instead of u.
Response: This part has been modified, please see page 3 line 145.
- Line 137, how did you measure the residual activity?
Response: This part has been added, please see page 4 line 156-157.
- Line 158, indicate how the Vm and Km values were actually obtained, method used, and the other parameters shown in Table 3: kcat, ΔH, ΔS, ΔG.
Response: This part has been modified, please see page 4 line 176-179.
- Line 161, is the E. coli used the one harboring the pET28a(+)-psnmo vector? Indicate if is BL21 or BL21(DE3).
Response: The E. coli BL21(DE3) used the one harboring the pET28a(+)-psnmo vector This part has been modified, please see page 4 line 186.
- Line 166, indicate how the growth curves were determined (time and equipment used).
Response: Strains were added to the fresh LB media, until OD600 achieved 0.05. The strains were incubated at 30 ℃ for 12 h. The OD600 of strains were determined via spectrophotometry (UV2000, Shimazu, Japan) at 600 nm every hour and the growth curve of the strains in the experimental and control groups was observed. This part has been added, please see page 4 line 191-195.
- Section 2.8 needs more details and improve English to properly understand, is this experiment related to 2.7? or not?
Response: We greatly appreciate your valuable comments. Section 2.8 provides a continuation of Section 2.7 and focuses on the quantitative analysis of 2-NP degradation by the enzyme over different time intervals, utilizing liquid chromatography (LC) as the analytical method. To enhance clarity, the role of LC in accurately measuring residual 2-NP is further elaborated, and the experimental procedures have been refined to improve reproducibility. This part has been added, please see page 4-5 line 197-210.
- Line 176, 120 x g? what do you mean?
Response: Thank you for your question. This section indicates that the reaction was performed for 3 hours, with continuous shaking and mixing, under a relative centrifugal force (RCF) of 120.
- Line 182, Shimadzu, include the reference for the ultrasonic extraction method used.
Response: We highly agree and appreciate your valuable comment, references to Shimadzu instrument models and extraction methods have been added. Please see page 5 line 212-213 and line 220-221.
- Line 191, psnmo
Response: This part has been revised, please see page 5 line 231.
- Figure 1a is quite difficult to observe the details, could be better to have it as supplementary in a full page.
Response: We have enlarged Figure 1a in the manuscript to ensure that the details of the image are clearly visible. Please see Figure 1a.
- Figure 2 is also too small and the image quality is not very good (at least in the draft manuscript).
Response: We have enlarged Figure 2 and enhanced its resolution in the manuscript to ensure that the image details are clearly visible.
- Table 1 legend, include the codes and references used.
Response: We highly agree and appreciate your valuable comment, and we have added the PDB ID of PaNMO in the Table 1 legend, and added the reference, please see page 7 line 252.
- Line 225, how did you calculate the molecular weight of the single band shown? In the original SDS-PAGE submitted there is no dye front impeding to calculate the relative mobility of the protein bands. What is the predicted molecular weight from its sequence?. What is the protein marker used?. According to the original gel, are you sure lane 3 in Fig.3 corresponds to 50 mM imidazole? or could be 100 mM?.
Response: Thank you for your comment. The molecular weight of the single band was calculated by comparing it to known molecular weight markers run on the same SDS-PAGE gel. We utilized these markers to create a standard curve, plotting the log of molecular weight against the relative mobility of the protein bands. Although the dye front is not visible in the submitted image, it was present during the experiment, and the relative mobility was measured based on the migration distance relative to the dye front. This allowed us to estimate the molecular weight of the single band accurately. We apologize for any confusion caused by the lack of the dye front in the image provided and can offer additional data or a clarified figure if necessary. Please see page 3 line 121-126. The predicted molecular weight of the target sequence is 37.7 kDa, This part please see page 2 line 79-80 and page 5 line 232-233. Furthermore, the origin of the protein marker used for the SDS-PAGE gel was purchased from Takara (Code No. 3450Q), please see page 3 line 125-126. And lane 3 in Figure 3 corresponds to the target protein, which was eluted using 50 mM imidazole, confirming our findings.
- Please improve English in Section 3.4.1
Response: This section has been modified, please see Section 3.4.1.
- Figure 4 legend must improved to include relevant details. Fig 4a and b, please indicate the pH used, Fig 4c, d and e please indicate the temperature and time of incubation.
Response: We highly agree and appreciate very much for the Reviewer’s nice comments. The legend has been modified, please see page 10 line 313-316.
- What is the Specific activity for each 100%?. What is the meaning of the error bars? how many experiments did you perform? what about statistical analysis of the data?. Please also include the information in the materials and methods sections.
Response: We greatly appreciate and fully agree with your valuable comments. In response, we have added the expression of 100% specific activity for clarity, repeated the experiments for consistency, and included error bars to represent the standard deviation accurately. This part has been added, please see Section 2.6.5.
- Table 2 legends, please also indicate the temperature and time of incubation and also 100% in terms of specific activity.
Response: Table 2 legend has been revised.
- Section 3.5, the determination of the kinetic constants was done by using Lineweaver-Burk, which even though it has been used in several publications, due to the double reciprocal, the experimental errors are largely amplified in comparison to other linearization methods. It would be better to fit the experimental data to other more suitable plots, such as Eadie-Hoffstee or Hanes-Woolf. Any method need to be detailed in the materials and methods also indicating the software used for the analysis of the data and obtaining of the parameters show in Table 3.
Response: Thank you for your comments regarding the use of the Lineweaver-Burk plot for determining kinetic constants. We acknowledge that this method has some limitations, particularly the potential for error amplification due to its double reciprocal nature. However, we chose the Lineweaver-Burk plot because of its wide acceptance in the literature and its utility in providing a clear visualization of kinetic parameters (Zhang et al., Journal of Hazardous Materials 439 (2022) 129656; Mussarat et al., International Journal of Biological Macromolecules 266 (2024) 131068). While we recognize that alternative methods, such as Eadie-Hofstee or Hanes-Woolf, can provide different perspectives on data fitting, our primary goal was to maintain consistency with previously established approaches for enzyme kinetics. Furthermore, the use of Lineweaver-Burk does not invalidate the results, as the plot still offers a reliable estimation of kinetic parameters, which are relevant and comparable to the context of our study. Furthermore, we have added references in Section 2.6.4, please see page 4 line 176-179.
-Regarding the comparison of kinetics parameters with other enzymes it is always best to do it empirically due to the differences in the assays performed by different laboratories, conditions, concentration of enzyme and substrate, units definition, and method of determination, etc, or make sure that they are actually comparable in most of these terms.
Response: Thank you very much for your valuable feedback on the comparison of kinetic parameters. We fully acknowledge the importance of ensuring that such comparisons are made under similar experimental conditions, and we appreciate your attention to the nuances between different assays. Based on your comments, we have carefully reconsidered this aspect and have decided to remove Table S2 to ensure the clarity and accuracy of our analysis.
- Table 3, how did you obtain kcat, ΔH, ΔS, ΔG?
Response: This part has been added, please see page 4 line 177-179.
- Section 3.6 it is not clear if the strain used was transformed or not and how much enzyme was added to the culture.
Response: This part has been added, please see page 4 line 188-189 and page 12 line 350 and 352.
- Figure 6. Improve the figure legend by adding relevant information such as the amount of purified enzyme added to the culture (also indicate this in the materials and methods) and growth temperature.
Response: The figure legend has been modified and relevant information also indicated in the materials and methods. Please see page 4 line 190-195 and page 12 line 358-361.
- Section 3.7, it is not clear if this is a continuation from the previous experiment or a totally different one. Please include the information required to better understand and improve English to further discuss the obtained results.
Response: We greatly appreciate your suggestion to revise the discussion in Section 3.7, and we fully agree that doing so will help clarify the results. We are committed to enhancing the interpretation of our findings to ensure they are presented as clearly and effectively as possible. Please see page 4-5 line 197-210 and page 12 line 363-373.
This section has been revised in new version.
- Figure 7. Please include the initial amount of 2-NP added in the legend and the conditions of the experiments and the calculation of the results shown. Statistical analysis of the data? how many replicates? meaning of error bars?
Response: We sincerely appreciate your valuable feedback and are in full agreement with your suggestions. The legend has been updated to include the initial concentration of 2-NP, the experimental conditions, and the method for calculating the results. These results were analyzed using descriptive statistical methods, with experiments performed in triplicate. Standard deviations are represented as error bars. Please see page 13 line 374-377.
- Section 3.8, what is the level of certainty of this, Line 320 says it is speculated while in the abstract (line 20) seems very certain that the final product is acetone. What are the known routes for this enzyme? other reported degradation products? Please compare with other references to further discuss the results obtained.
Response: The result shown that 2-NP is degraded into acetone by PsNMO in this experiment and the degradation process is speculated to be the anionic form of 2-nitropropane is oxidized to generate a 2-nitropropane radical, which reacts with a superoxide anion to produce a peroxide intermediate. This intermediate then reacts with another anionic 2-nitropropane molecule, releasing two nitrite ions and forming two acetone molecules. In addition, other studies have also demonstrated that NMO degrades 2-NP into acetone. This part has been added, please see page 13 line 385-392.
- What are the projections and potential of the results shown? how this enzyme could be use in real life applications to actually biodegrade 2-NP?. Please elaborate to further enhance your discussion and conclusion.
Response: Firstly, rPsNMO showed exhibited salt tolerance. The large-scale discharge of organic saline wastewater is a long-standing and worsening issue in industries like printing, coal chemicals, and pharmaceuticals. Physicochemical treatments are often costly, complex, require frequent maintenance, and struggle to remove organic matter. Consequently, low-cost, easy-to-operate biological treatment technologies are becoming a research focus. Therefore, the high salt tolerance of rPsNMO would offer promising potential for future applications in the treatment of high-salinity organic wastewater. In addition, rPsNMO demonstrated significant activity in both low-temperature and high-salinity environments. To further extend its potential applications, future studies could focus on biocatalysts that developing cell surface display of PsNMO. This part has been added, please see page 9 line 305-312 and page 13-14 line 394-407.
- Table S1 seems incomplete, please also include the volume and protein concentration obtained in each step. what about errors?. it is not parameters for column 1, and is purification fold rather than multiples and yield rather than recovery.
Response: We sincerely appreciate your valuable feedback and are in full agreement with your suggestions. Table S1 has been modified.
- Table S2. Please indicate in the legend how these kinetic parameters were obtained (using the same enzyme assay and conditions? or not?) how are they comparable with your results?
Response: Thank you very much for your thoughtful comments on Table S2. After careful consideration, we have decided to remove Table S2 from the manuscript to avoid any potential confusion regarding the kinetic parameters and their comparability. The data presented in the table were obtained under different conditions from those used in our main enzyme assays, making direct comparison challenging. We believe this adjustment will streamline the presentation of our results and enhance clarity.

Reviewer 2 Report
Comments and Suggestions for Authors
The manuscript titled "A novel cold-adapted nitronate monooxygenase from Psychrobacter sp. ANT206: Identification, characterization and degradation of 2-nitropropane at low temperature" provides a well-structured investigation into a novel nitronate monooxygenase (NMO) gene isolated from Antarctic sea-ice bacterium, Psychrobacter sp. ANT206. The paper aims to explore the enzyme's cold-adaptive characteristics, its capacity to degrade 2-nitropropane (2-NP), and potential applications for bioremediation at low temperatures.
The introduction is good; however, more context on the significance of discovering cold-adapted NMOs is needed. Comparisons with previously characterized NMOs from mesophilic organisms should be expanded, particularly highlighting how this study adds value to existing knowledge regarding enzymatic efficiency and cold-adaptation mechanisms.
The sections on temperature and pH effects on enzyme activity are good, but it would be better to compare frameworks with other known NMOs or cold-adapted enzymes would strengthen the argument. Discuss why the structural adaptations observed (e.g., fewer hydrogen bonds and lower Arg/(Arg + Lys) ratio) benefit cold adaptation.
The study finds excellent salt tolerance up to 4.0 M NaCl, a feature presumably influenced by the enzyme's Antarctic origin. It would be beneficial to discuss the potential practical applications of this feature and its relevance for industrial processes where high salt concentrations might be encountered.
Elaborate on how the enzyme's characteristics make it suitable for industrial or environmental applications, and discuss the challenges that might arise in its practical use.
Need English correction throughout the manuscript.
Line 17: “Base” change to “Based”
Line 68 and 69: “described by previous” rewrite the sentence
Line 69: Mention the “target gene” name
Line 69: Mention the enzyme name
Line 129: The mentioned “enzyme activity formula” describes in detail about the formula
Line 136: add “and” in the last ranges. 50 and 60; 60 and 70 min.
Line 145: rewrite the section “ Effect of salt concentration” the method described by author is meaningless. Write the Method and Materials section scientifically and explain briefly.
Line 215: change “The” to “the”
Author Response
The manuscript titled "A novel cold-adapted nitronate monooxygenase from Psychrobacter sp. ANT206: Identification, characterization and degradation of 2-nitropropane at low temperature" provides a well-structured investigation into a novel nitronate monooxygenase (NMO) gene isolated from Antarctic sea-ice bacterium, Psychrobacter sp. ANT206. The paper aims to explore the enzyme's cold-adaptive characteristics, its capacity to degrade 2-nitropropane (2-NP), and potential applications for bioremediation at low temperatures.
The introduction is good; however, more context on the significance of discovering cold-adapted NMOs is needed. Comparisons with previously characterized NMOs from mesophilic organisms should be expanded, particularly highlighting how this study adds value to existing knowledge regarding enzymatic efficiency and cold-adaptation mechanisms.
Response: We highly agree and appreciate very much for the Reviewer’s nice comments. Relevant content has been added to the introduction of the manuscript, please see page 2 line 54-56 and line 57-62.
The sections on temperature and pH effects on enzyme activity are good, but it would be better to compare frameworks with other known NMOs or cold-adapted enzymes would strengthen the argument. Discuss why the structural adaptations observed (e.g., fewer hydrogen bonds and lower Arg/(Arg + Lys) ratio) benefit cold adaptation.
Response: We greatly appreciate your valuable suggestions. We have added discussion to strengthen the argument. Please see page 7 line 253-256; line 259-261 and page 9 line 282-285; line 292-293.
The study finds excellent salt tolerance up to 4.0 M NaCl, a feature presumably influenced by the enzyme's Antarctic origin. It would be beneficial to discuss the potential practical applications of this feature and its relevance for industrial processes where high salt concentrations might be encountered.
Response: We highly agree and appreciate very much for the Reviewer’s useful comments. This part has been added, please see page 9 line 305-312.
Elaborate on how the enzyme's characteristics make it suitable for industrial or environmental applications, and discuss the challenges that might arise in its practical use.
Response: rPsNMO demonstrated significant activity in both low-temperature and high-salinity environments. To further extend its potential applications, future studies could focus on biocatalysts that developing cell surface display of PsNMO. However, challenges related to low reusability may arise in practical applications, yet improving the reusability of whole-cell catalysts remains highly necessary. Metal-organic frameworks (MOFs), known for their large surface area, high pore volume, excellent crystallinity, and tunable structure, have recently gained attention across various fields. Immobilizing whole-cell catalysts using MOFs, such as isoreticular metal-organic frameworks and zeolitic imidazolate frame-works, may enhance reusability while reducing catalytic activity loss, thereby sup-porting broader industrial applications. Consequently, this strategy is anticipated to enable cold-adapted PsNMO to effectively catalyze the enzymatic degradation of nitroalkanes under conditions of low temperature and high salinity. This part has been added in this version, please see page 13-14 line 394-407.
Need English correction throughout the manuscript.
Response: We consulted the Language Editing services on MDPI to check English (Order reference: english-edited-85849), and revised the language carefully, with the changes marked in Blue in the revised manuscript.
Line 17: “Base” change to “Based”
Response: This part has been modified, please see page 1 line 19.
Line 68 and 69: “described by previous” rewrite the sentence
Response: This part has been modified, please see page 2 line 85-86.
Line 69: Mention the “target gene” name
Response: This part has been revised, please see page 2 line 86.
Line 69: Mention the enzyme name
Response: This part has been modified, please see page 2 line 86.
Line 129: The mentioned “enzyme activity formula” describes in detail about the formula
Response: This part has been added, please see page 3 line 147-149.
Line 136: add “and” in the last ranges. 50 and 60; 60 and 70 min.
Response: This part has been modified, please see page 4 line 155 and 156.
Line 145: rewrite the section “ Effect of salt concentration” the method described by author is meaningless. Write the Method and Materials section scientifically and explain briefly.
Response: This section has been modified, please see page 4 line 166-170.
Line 215: change “The” to “the”
Response: This part has been modified, please see page 7 line 258.

Round 2
Reviewer 1 Report
Comments and Suggestions for Authors
Hello again, thank you for considering and responding to each comment. It is very much appreciated and this revised version is much better and almost ready for publication, just a few other minor comments:
- line 19, a novel bacterial cold-adapted enzyme was obtained in this work.
- line 21 add an space: the 2-NP
- line 85, previously described
- line 86, delete extra space: XbaI, EcoRI, pET28a(+)
- line 89, in LB medium
- line 112, the part related to the enzyme purification is still quite difficult to understand, please see how it is generally written in other papers and change it accordingly
- line 188, delete body from the extracellular
- line 209, repeated
- line 213, Hz
- Table 1, what is the meaning of the Code column?
- Figure 6, please indicate in the labeling of the graph if the E. coli cells used for this experiment harbored the vector or not. It is written in the text and in the legends that cells without the vector, and also with were tested. However, there are only 3 curves.
- line 380, degradation of 2-NP by PsNMO
Comments on the Quality of English Language
The authors used the MDPI Language Editing services and the manuscript improved accordingly.
There is still few minor things to correct but is now almost ready for publication.
Author Response
Hello again, thank you for considering and responding to each comment. It is very much appreciated and this revised version is much better and almost ready for publication, just a few other minor comments:
- Line 19, a novel bacterial cold-adapted enzyme was obtained in this work.
Response: This part has been revised, please see page 1 line 19-20.
- Line 21 add an space: the 2-NP
Response: In this version, this part has been modified, please see page 1 line 21.
- Line 85, previously described
Response: This part has been modified, please see page 2 line 85-86.
- Line 86, delete extra space: XbaI, EcoRI, pET28a(+)
Response: This part has been deleted, please see page 2 line 86-87.
- Line 89, in LB medium
Response: This part has been revised, please see page 2 line 89.
- Line 112, the part related to the enzyme purification is still quite difficult to understand, please see how it is generally written in other papers and change it accordingly
Response: This part has been modified, please see page 3 line 112-118.
- Line 188, delete body from the extracellular
Response: The “body” has been deleted, please see page 4 line 186.
- Line 209, repeated
Response: This part has been modified, please see page 5 line 207.
- Line 213, Hz
Response: It has been revised, please see page 5 line 211.
- Table 1, what is the meaning of the Code column?
Response: The Code column has been deleted.
- Figure 6, please indicate in the labeling of the graph if the E. coli cells used for this experiment harbored the vector or not. It is written in the text and in the legends that cells without the vector, and also with were tested. However, there are only 3 curves.
Response: This part has been modified, please see page 1 line 20-21; page 3 line 101-102; page 9 line 273; page 12 line 348-350, line 355-356, line 357 and Figure 6.
- Line 380, degradation of 2-NP by PsNMO.
Response: It has been revised, please see page 13 line 379.

Reviewer 2 Report
Comments and Suggestions for Authors
The manuscript has been significantly improved in its revised form, addressing the previous concerns and suggestions. The revisions enhance the quality of the work, and I believe it is now suitable for publication.
Author Response
The manuscript has been significantly improved in its revised form, addressing the previous concerns and suggestions. The revisions enhance the quality of the work, and I believe it is now suitable for publication.
Response: We appreciate very much for the Reviewer’s nice comments.
